

# Unveiling the urban colonization of the Asian water monitor (*Varanus salvator*) across its distribution range using citizen science

Álvaro Luna[1] and Armand Rausell-Moreno[2]

[1] Department of Biosciences, Universidad Europea de Madrid, Madrid, Madrid, Spain
[2] Department of Biogeography and Global Change, National Museum of Natural Sciences (MNCN-CSIC), Madrid, Madrid, Spain

## ABSTRACT

**Background**. This study aims to investigate the urban colonization of the Asian water monitor (*Varanus salvator*) across its entire range of distribution, addressing the paucity of research on this species in urban ecosystems. The research spans the geographic range of the Asian water monitor, focusing on urbanized areas where the species accumulates more observations (Bangkok, Colombo, Jakarta, Kuala Lumpur and Singapore).

**Methods**. We conducted a systematic review to comprehensively assess the current knowledge of the species' presence in cities. Additionally, citizen science data from repositories like GBIF (Global Biodiversity Information facility) were utilized to analyze the distribution patterns of *V. salvator* in urban environments. To elucidate urban distribution and correct collection biases, observations were weighted by sampling effort, using as a proxy all squamate occurrences available from 2010–2023, including *V. salvator*.

**Results**. Despite the widespread presence of the Asian water monitor in numerous cities within its distribution range, the available studies on the topic appear to be scarce. Existing research primarily consists of descriptive reports on diet and behavior. Our findings indicate that *V. salvator* predominantly colonizes green patches in urban areas, such as parks and small gardens. Larger cities exhibit higher records, potentially due to both permanent populations and increased citizen science reporting.

**Conclusions**. The Asian water monitor, as the largest lizard with established populations in cities, remains scarcely studied on a broader scale. However, the urban design of each city seems relevant to understand the distribution patterns within each context. Our study highlights the need for further research to explore the ecological and human dimensions associated with the species' presence in urban environments.

Corresponding author
Álvaro Luna,
alvaro.luna@universidadeuropea.es

# INTRODUCTION

Urbanization is one of the most important drivers of habitat transformation, accelerated in recent decades due to the growth of the human population, especially in Africa and
Asia (*Seto, Güneralp & Hutyra, 2012*; *Johnson & Munshi-South, 2017*). Thriving in urban environments can lead to new ecological challenges and selection pressures for wildlife, for example influencing the dispersal decisions (*Marzluff et al., 2016*; *Evans et al., 2017*; *Luna et al., 2019*), the reproductive strategies and success (*Seress et al., 2020*; *Luna et al., 2021*; *Saulnier et al., 2023*) and the dietary habits (*Galbraith et al., 2015*; *Teyssier et al., 2020*), but also exposing individuals to different predatory pressures (*Shwartz et al., 2009*; *Eötvös, Magura & Lövei, 2018*) and parasite- host interactions (*Delgado-V & French, 2012*; *Sáez-Ventura et al., 2022*). However, although the sprawl of cities usually results in the simplification and homogenization of animal communities through local extinction processes (*Sol et al., 2014*), some species successfully exploit urban environments, that often support higher population densities and reproductive rates compared to rural areas (*Rebolo-Ifrán, Tella & Carrete, 2017*), with examples even in endemic and threatened species (*Vignoli et al., 2009*; *Luna et al., 2018*; *Woolley et al., 2019*).

Although urban ecological science is now a widely recognized field within ecology, some major gaps still remain unresolved (*Shochat, Warren & Faeth, 2006*). Thus, to date, most of the research is focused on single cities located in Europe, North America and Australia, with a significant preponderance of studies about plants, birds and mammals (*Magle et al., 2012*; *Rega-Brodsky et al., 2022*). Regarding urban reptiles, a review conducted by *Brum et al. (2023)* confirmed that some biases still persist and the current knowledge is unbalanced, with developing tropical and megadiverse countries often overlooked in comparison to studies about reptiles inhabiting cities of temperate areas. This study also shows how, within reptiles, research focused on the order Squamata is underrepresented, as is also the case with those of *Lacertilia* among this order. In general, the sprawl of cities is considered a threat to reptiles worldwide (*Cox et al., 2022*), due to the loss of habitat for feeding, breeding and shelter, the urban pollution, the increasing number of paved roads and the impact of domestic animals such as dogs and cats (*White & Burgin, 2004*; *Croteau et al., 2008*; *Perry et al., 2008*; *Cordier et al., 2021*). However, reptiles show varied responses to urbanization, influenced by factors such as species life-history, dispersal strategies, habitat availability, key resources, and patch connectivity (*Garden et al., 2007*; *Hamer & McDonnell, 2010*), with consequences as a limited genetic flow, alterations in endocrine stress responses in comparison to rural counterparts, and different exploratory and foraging behaviors, reduced risk perception and response rates to predators and human presence (*French et al., 2018*). As a result, some species successfully thrive in urban environments (*Ackley et al., 2015*; *Davis & Doherty, 2015*; *Entiauspe-Neto, Perleberg & De Freitas, 2016*). For example, *Turak et al. (2020)* explored the presence of reptiles related to freshwater ecosystems within 50 km of cities at global level, using records from online databases like GBIF (Global Biodiversity Information Facility), and revealed how several species are sighted in or near many cities with more than 100,000 inhabitants.

As an increasing majority of humans will reside in cities in coming years (70% of the human population is expected to live in cities by 2050; *United Nations, 2018*), scientists, conservationists, and politicians highlight that a better understanding of the patterns that explain the biodiversity of cities and recognizing the value of their conservation is a key challenge for the next decades (*Dearborn & Kark, 2010*; *Shwartz et al., 2014*; *Threlfall et*

*al., 2015*). In this sense, citizen science offers new opportunities for researchers, being a cost-effective method for collecting valuable ecological data through public participation (*Cooper et al., 2007*). As many people live in urban areas, projects involving citizen science can have special success in cities, helping to monitor populations and detect new species (*McCaffrey, 2005*; *Anton et al., 2018*; *Roger & Motion, 2022*). For example, the increasing popularity and data validation of online platforms like eBird and iNaturalist (*Sullivan et al., 2014*; *Beninde et al., 2023*) provide millions of digital observations submitted by global users, most of them in cities (*Kelling et al., 2015*). Here we explore the presence of the Asian water monitor (*Varanus salvator)* in cities along its natural distribution, that includes the Indian subcontinent and Southeast Asia. For our purpose, we first review the scientific literature available for this species, gathering information about how many studies are conducted in urban habitats in proportion to other habitats, and the topic studied in the articles analyzed when they are developed in cities. Moreover, to explore where and how the species is occurring in cities, we use citizen science data obtained from GBIF, focusing on certain urban areas where most of the records of Asian water monitors are concentrated. From this second approach, we analyze the location and distribution of the species in different city border-center gradients to understand the urban patches they preferentially exploit. Moreover, we also explore potential biases related to the citizen-science data and the importance of considering potential constraints while developing comprehensive approaches. By knowing the potential and limitations of this data, it can be extremely useful to assess species' distribution for conservation purposes.

## MATERIAL AND METHODS

### Study species and area

The Asian water monitor (*Varanus salvator*, Laurenti 1768; order Squamata; family Varanidae) has six subspecies recognized: *Varanus s. salvator*, *V. salvator andamanensis*, *V. salvator bivittatus*, *V. salvator celebensis*, *V. salvator macromaculatus* and *V. salvator ziegleri* (*Auliya, 2006*; *Quah et al., 2021*; *Auliya & Koch, 2020*).  It is one of the most widely distributed varanids, ranging from Sri Lanka in the west, to the Celebes Islands (Indonesia) to the east and South China to the north, occurring mostly in Southeast Asia countries (*Gaulke & Horn, 2004*; *Bennett et al., 2010*; *Quah et al., 2021*). The Asian water monitor is among the biggest lizards in the world, reaching almost 3 m from head to tail. Males mature at a smaller size but grow to reach larger body sizes (*Shine & Harlow, 1998*; *Frýdlová et al., 2011*) and have longer tails. Observations made in Bangkok (Thailand) show how they have bimodal diurnal activity, hunting/scavenging in the morning (06:00–08:00 h) and the afternoon (15:00–17:00 h), spending the rest of the day basking and floating (*Trivalairat & Srikosamatara, 2023*). This diurnal activity was also confirmed in the Sundarbans (Bangladesh) by *Rahman, Rakhimov & Khan (2017)*. Considering data from Sumatra, this species extend egg-laying season from April to October, with the possibility to produce more than one clutch each per year, ranging from five to approximately 20 eggs (*Shine, Harlow & Keogh, 1996*). They are able to thrive in both terrestrial and aquatic environments (*Zhao, Zhao & Zhou, 1999*; *Gaulke & Horn, 2004*; *Weijola, 2010*; *Weijola & Sweet, 2010*;

*Bennett et al., 2010*), including highly human-altered landscapes such as farmlands and cities (*Cota, Chan-Ard & Makchai, 2009*). They have a very flexible diet, that includes invertebrates, eggs, fish and even carrion (*Bennett et al., 2010*; *Grismer, 2011*; *Rahman, Rakhimov & Khan, 2017*; *Yu et al., 2021*). *V. salvator* has been intensively harvested for the skin industry (*Shine, Harlow & Keogh, 1996*; *Traeholt, 1998*; *Khadiejah et al., 2019*), the traditional medicine (*Mardiastuti et al., 2021*) and also for its consumption as food (*Arida et al., 2021*).

## Literature review

To review the scientific literature available about the Asian water monitor we followed and adapted the guidelines proposed by *Haddaway et al. (2015)*. The main steps used for our study are summarized in a flow diagram in the supplementary material (SP1). Briefly, first we studied peer-reviewed articles published in journals available on Scopus and Web of Science databases. The search was applied to the title, including any dates. We also included an additional non-systematic search using Google Scholar (*Gehanno, Rollin & Darmoni, 2013*; *Piasecki, Waligora & Dranseika, 2018*). Moreover, we used the "snowball" procedure, including those articles related to our topic found in the selected and analyzed references (*Lozano et al., 2019*). Initially we considered the inclusion of the so-called grey literature, attending to the additional contribution of technical reports and other sources beyond scientific articles (*Haddaway & Bayliss, 2015*). We only considered articles written in English, and we discarded those articles without an online version and also those where the link provided and parallel searches did not lead to the referred article. The review was conducted using a search with the term "*Varanus salvator*" AND "urban environment", but previously we used other potential combinations that did not yield adequate results for the objective of the work (*i.e.*: articles not related to *Varanus* lizards, articles in urban ecosystems but focused on human well-being and other aspects related to urban life of humans, etcetera). Regarding the topic in the studies we reviewed, in our final list we discarded those studies based on reviews and theoretical/conceptual articles without their own data and analyses. Lastly, to avoid the heterogeneity of inclusion criteria inherent to different observers, only one of the authors carried out the search and the subsequent exploration of the articles.

We revised the content of the retained articles by a two-step process. First, we screened titles and abstracts of the first 200 articles obtained in our search, ordered by relevance and without limiting to any date. We do not consider as 'urban' those articles conducted in human modified landscapes such as agricultural areas but not considered properly cities. Similarly, articles focused on the use of *V. salvator* in laboratory conditions for medical research were discarded. Secondly, we read the main text of the articles to gather basic information to separate the studies conducted in urban environments, and to obtain information regarding the year of publication, the topic addressed and the country in which the study was conducted. We classified the articles retained according to defined categories: behavior, diet, physiology, distribution, habitat selection, parasitology, and conservation (see details in SP2). A given article can be included in more than one category, if it focuses on more than one aspect according to our classification.
## GBIF data collection and analyses

We acquired occurrence data of Asian water monitors from the Global Biodiversity Information Facility (GBIF) (http://www.gbif.org), one of the largest sources of open data, that includes information from different citizen science online platforms, hence being of great interest to ecologists working on the spatial distribution of species (*Telenius, 2011*; *Beck et al., 2014*; *Ivanova & Shashkov, 2021*). As a first step, we downloaded 4,584 occurrences (*Gbif.org, 2022*). We then filtered by year (>2010) and verified veracity of data one by one by spatial confirmation using QGIS (v. 3.16) (*i.e.,* avoiding those records that for any reason appeared on the sea instead of on terrestrial ecosystems). We also checked the linked image (we only considered data with pictures) from iNaturalist (GBIF uses observations from iNaturalist as well) associated with each occurrence, discarding, when necessary, any doubtful data (for example when the individual in the image is in captivity or when the record was not correctly identified and could refer to other *Varanus* species). We only kept data within urbanized areas from all the distribution range of the species.

As a second step, for subsequent analysis we selected the top 5 cities with more records. For those cities, we delimited urban areas as raster cells with a value higher than 20 for Human Footprint (HFP) index (*Venter et al., 2018*), as a clear boundary could be detected between less anthropized environments surrounding the city and the urban limits at this threshold. We chose this layer because of its common usage (*Santini et al., 2021*) and its great potential to indicate human influence, quantifying the impacts of human disturbance on a global scale. Since observations can have spatial biases with proven consequences on species distribution modelling and spatial analysis (*Hortal, 2008*), we sought to reduce them by calculating the ratio of observation, therefore, avoiding working with occurrence abundance. In order to gather enough information about sampling efforts, this ratio used all squamate occurrences available for the study area from 2010–2023 as a proxy (*Gbif.org, 2023*), including *V. salvator*, and the observations were divided by the number of different observers (Squamate observations/N° Observers) per pixel ($\approx$1 km$^2$) (*Chauvier et al., 2021*). By doing this, we obtained an estimation of how the sampling effort is distributed along our cities in order to support if the accumulation of water monitor observations could correspond to real abundance. We selected squamates because they are prevalent in our study areas and they are the taxonomic group of our study species. The observation patterns could differ between the different types of squamates and those of *V. salvator*. However, the *V. salvator* records are not abundant enough to calculate an observation ratio of its own, so the comprehensive records within the entire taxonomic group (squamates) should adequately represent the distribution of sampling effort across the cities for this particular group. Afterwards, we calculated the distance of those pixels with at least one *V. salvator* observation to the urban border defined by the HFP, which allowed us to explore possible patterns of distribution inside cities.

Additionally, we analyzed the approximated location where these observations occur within the urban matrix. For this aim, we preliminary checked if we could determine patterns that reveal greater sampling efforts, and therefore, higher *V. salvator* presence in urban green areas (parks and big gardens) in comparison to other urban environments such as neighborhoods, small gardens, natural and artificial streams, *etc*. For this, we determined

 

as "Park" those areas with a Vegetation Continuous Field (VCF) value higher than 7, as in the previous case, it seemed to be a natural break to distinguish urban from green areas. This last raster layer is global representation of surface vegetation cover (*DiMiceli et al., 2015*) and was generated using Google Earth Engine derived from a 250 m resolution from MODIS. In both cases, HFP and VCF, there was a pixel size of 0.00833° (∼1 km$^2$ at the equator).

## RESULTS

### Literature review

Our search was conducted on 8th April 2021. We retained 148 scientific articles after the steps applied to discard or include them (see SP3 to see a detailed list of the articles reviewed). Most of the articles (96) were found in Scopus, Web of Science or in both sources. Moreover, we added 49 scientific articles found in Google Scholar and three articles detected by snowball. Although we initially considered grey literature, we did not find technical reports, conferences and similar publications, so we only included scientific articles. Most of the articles (117) were published after the 2000s, and only 31 articles before. There are 13 countries represented in those studies (Bangladesh, China, India, Indonesia, Laos, Malaysia, Myanmar, Philippines, Singapore, Sri Lanka, Thailand, Timor-Leste, Vietnam). However, in 46 cases the country was not described, or the study was conducted in laboratory conditions (*i.e.,* the species was not studied in its habitat). Considering the total of studies reviewed, only 17 articles focus in urban habitats. We found that the studies of urban Asian water monitors started around the 2000s, and between 0–2 articles focused on urban populations of this species are published yearly to date. Moreover, we detect that all the studies analyzed in urban habitats were conducted in three countries: 12 in Thailand, three in Sri Lanka and two in Indonesia (Fig. 1A). The predominant topics of those studies were the behavior of these reptiles in cities, their diet, and to a lesser extent the distribution of the species, while studies focused on physiological aspects, parasitology, habitat selection and the conservation of the species are only represented with one study (Fig. 1B).

### Urban occurrence using GBIF data

We show how *V. salvator* occurs in Asian urbanized areas from Sri Lanka in the west to East Indonesia, with most records concentrated in cities of Thailand, Sri Lanka, Singapore, peninsular Malaysia, Java and Sumatra (Fig. 2A). Using this source, we did not find data from Vietnam, Laos, Cambodia, Timor Leste, Philippines and China. Looking more closely to our data, we detect the five cities with the most records: Colombo ($n = 36$), Kuala Lumpur ($n = 99$), Jakarta ($n = 42$), Bangkok ($n = 481$), Singapore ($n = 1,641$) (Fig. 2B). For Colombo and Jakarta we observe that sampling effort (*i.e.,* more observations per observer) tends to accumulate near the city center (Fig. 3). Singapore, Kuala Lumpur and to a lesser extent Bangkok, show a balanced distribution of sampling effort with a slight increase towards the centered areas mainly for the first two. Bangkok, however, shows a maximum observation ratio at a distance of 6.25 km from the border. These observation spots (1 km$^2$ pixels with at least one *V. salvator* observation) are not equally distributed
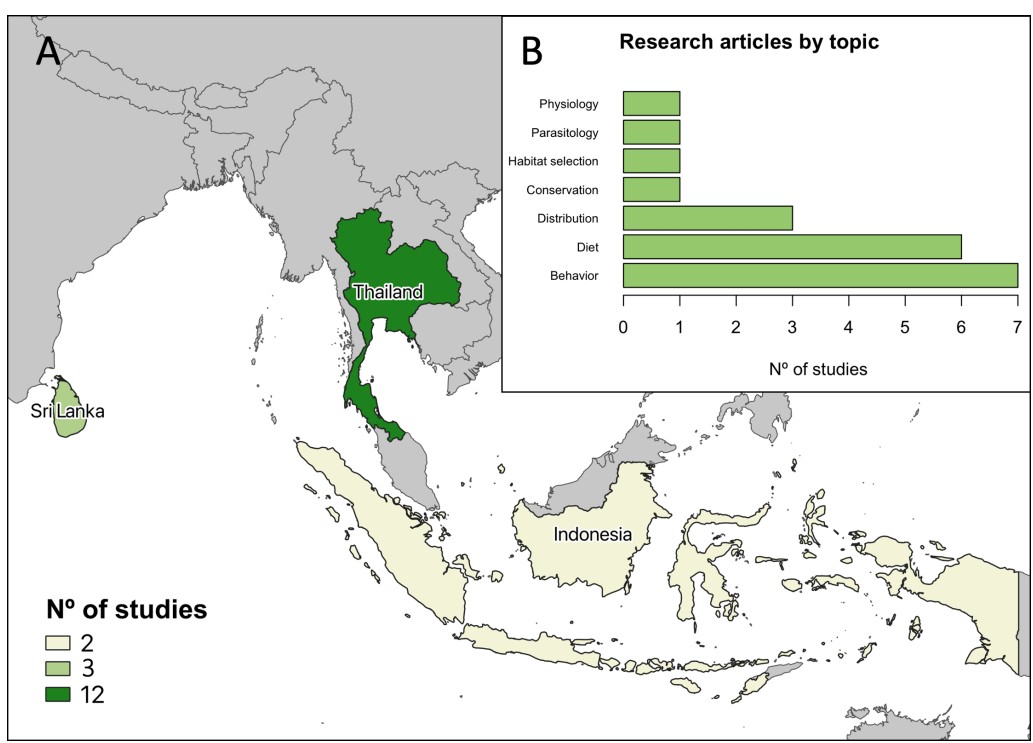

**Figure 1  Main results of the literature review.** (A) Number of studies per country found about *Varanus salvator* in urban environments; darker colors represent high number of articles found for a given country. (B) Classification by topic studied in the articles reviewed (each article can be included in more than one category). Figure made by QGIS version 3.32.3.

either (Fig. 3). Thus, mainly for Colombo and Jakarta we see gaps on the distribution of black dots along the distance to border axis, which indicates uneven observation patterns or incomplete sampling within the city. Bangkok, Kuala Lumpur and Singapore show a more constant representation of observations along the border to city-center gradient.

We also studied the distribution of records within the urban matrix and how this sampling effort is distributed among habitat types. Colombo, Jakarta and Bangkok experience at least some degree of net positive increase in the intensity of sampling when increasing VCF (Fig. 4). In particular, Jakarta experiences a high point at around a value of 7.25, near the stablished threshold for being considered as park (VCF > 7). Kuala Lumpur reaches its peak around a value of 12.25 for VCF. After that, the effort decreases. Singapore, however, again shows a homogenous pattern of sampling effort when it comes to different types of habitats. Colombo, Jakarta and Kuala Lumpur presented some data gaps for the VCF variable (see Fig. 4), while in Singapore and Bangkok the data are better distributed. We observed that spotting areas of *V. salvator* occur mainly in green urbanized environments: in Colombo (87.5%), Jakarta (77.4%), Kuala Lumpur (53.1%) and Singapore (76%) a great percentage of the spotting areas occur in to these defined park areas. Only in Bangkok, (26.9%) we observed a greater number of observations out of parks (Fig. 4).

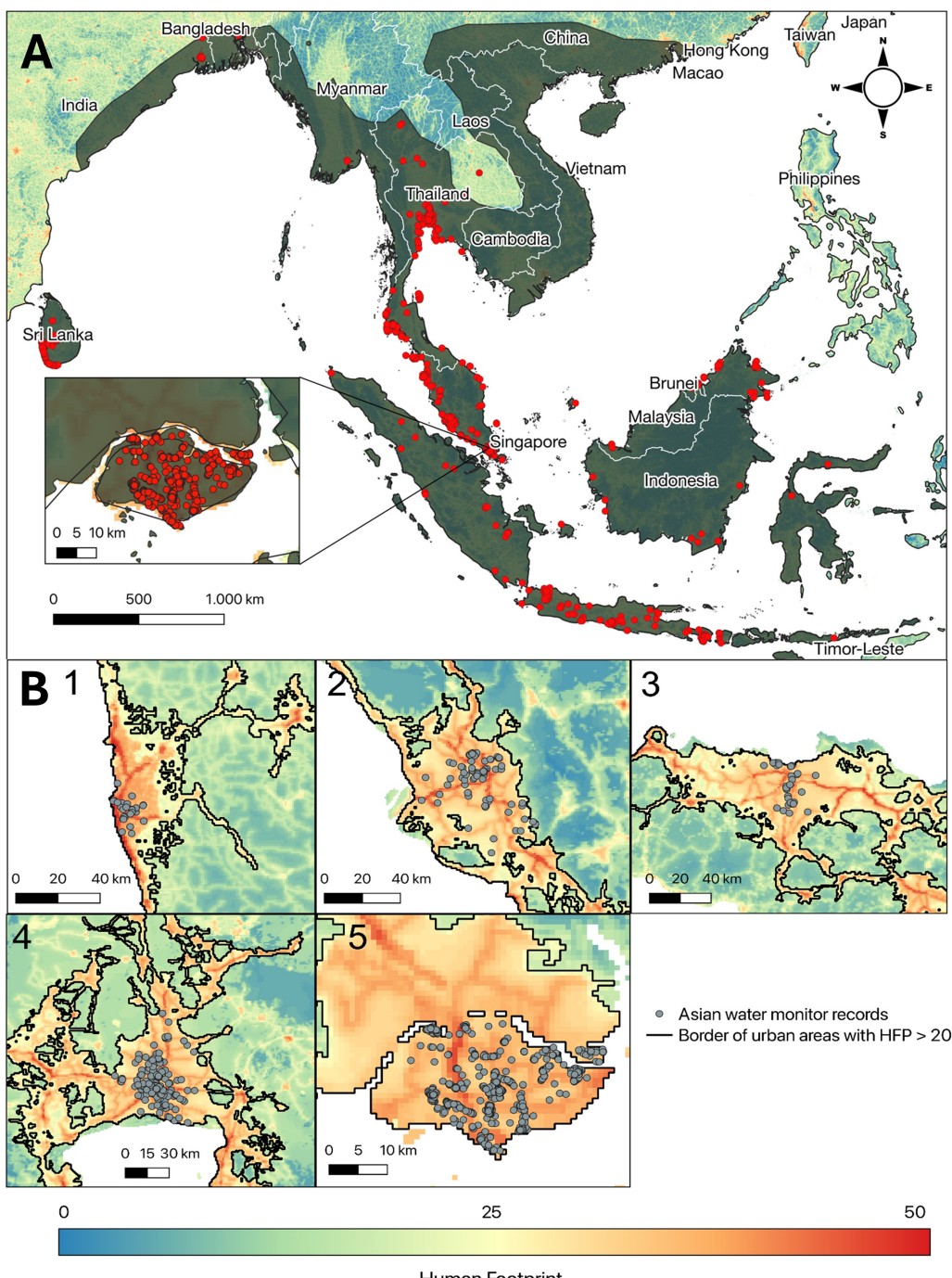

**Figure 2  Records of urban monitor lizards (*Varanus salvator*) in its native range.** (A) Distribution range of the Asian water monitor (*Varanus salvator*) in grey. The red dots represent urban records of *V. salvator* obtained from GBIF. (B) Five cities with the most Asian water monitor records (in grey) (1: Colombo, 2: Kuala Lumpur, 3: Jakarta, 4: Bangkok, 5 Singapore). Color gradient represents Human Footprint, red being the highest values and blue the lowest ones. The black line separates urban areas with a Human Footprint value higher than 20 from less urbanized environments. Figure made using QGIS version 3.32.3.

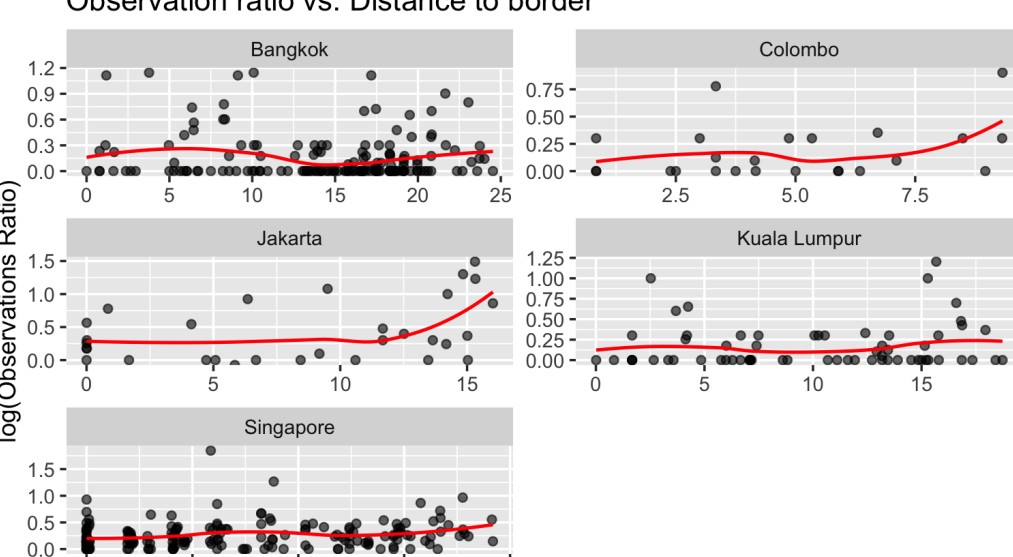

**Figure 3  Records of urban *Varanus salvator* attending to the distance to the city border.** Logarithmic distribution of sampling effort (observation ratio) within *V. salvator* observation areas along a distance gradient in kilometers. The red line shows tendency of the sampling effort while black dots represent observation areas (each pixel) with at least one *V. salvator* occurrence. 0 represents points in the border.

## DISCUSSION

### Literature review

Our results confirm the apparent lack of scientific literature on the colonization and ecology of Asian water monitors (*V. salvator*) in urban environments, with only 17 of the 148 articles reviewed conducted in urban environments, and few cities of three countries (Indonesia, Sri Lanka and Thailand) frequently represented in those studies. Most of the articles reviewed are short notes and reports of field observations, mostly about behavior, diet and the how the species thrive in this habitat. Those records confirm the generalist feeding habits of the species, which in cities consumes diverse birds, mammals, fish, amphibians and reptiles (*Bundhitwongrut et al., 2008*; *Karunarathna, Amarasinghe & De Vos, 2008*; *Stanner, 2010*; *Cota & Sommerlad, 2013*; *Mahaprom & Kulabtong, 2018*), but also exploit carrion and are attracted to human waste (*Kulabtong & Mahaprom, 2015*; *Lawton et al., 2008*). Regarding the behavior, the studies reviewed describe the mating of the species, the intraspecific relations in different seasons, the use of underwater burrows, its daily activity patterns, with both diurnal and nocturnal habits, and how the Asian water monitors preferentially use aquatic habitats, especially the hatchlings and juveniles (*Rathnayake et al., 2003*; *Cota, 2011a*; *Cota, 2011b*; *Karunarathna et al., 2017*). In this sense, the urban environment facilitates observations of basic ecological and behavioral aspects that could be more difficult to observe in other contexts, mostly due to the more elusive behavior of non-urban individuals of many species, compared in some studies in both

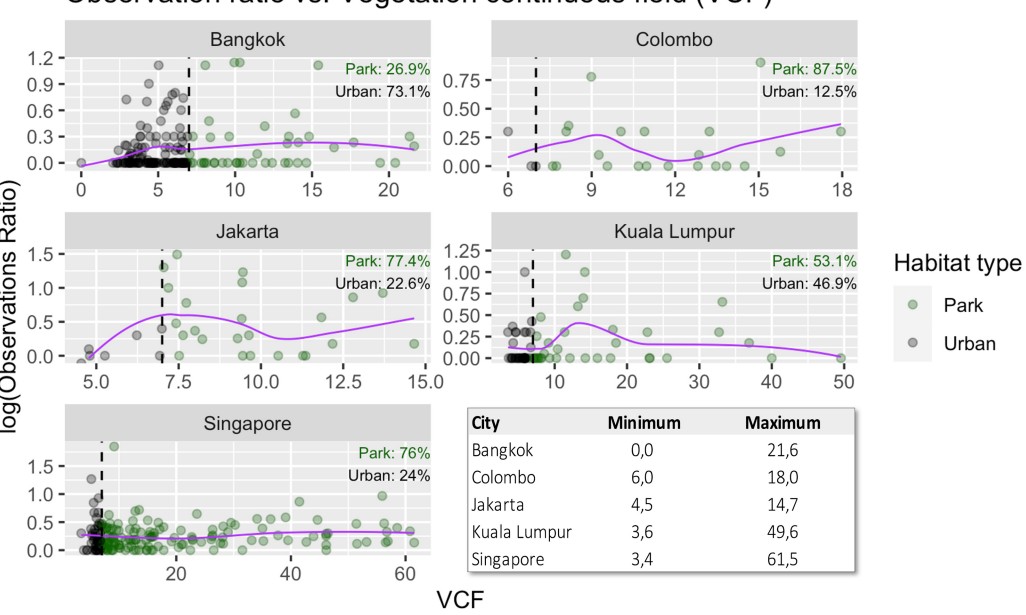

**Figure 4** Records of *Varanus salvator* in cities attending to the greenery level of the urbanized area. Logarithmic distribution of sampling effort (measured as observation ratio) between parks and the urban matrix. Red line represents the tendency of sampling effort with the increase of VCF while dots represent observation areas (1 km pixel) of Asian water monitors for the two different types of habitats studied. Green: park; Grey: urban. Percentages show the proportion of sampling areas that belong to each category. The table displays minimum and maximum values of VCF for each city.

urban and non-urban environments (*Evans, Boudreau & Hyman, 2010*; *Isaksson, AD & Gil, 2018*). Nevertheless, it is possible that our findings may be biased due to the small sample size obtained through the revision, together with the apparent absence of studies offering comparisons across a broader geographical scope or studies that articulate their hypotheses within the theoretical framework of urban ecology.

## Urban occurrence using GBIF data

Attending to our results, *V. salvator* is colonizing cities in different countries within its distribution range, with differences regarding how they exploit the urban matrix among cities. This is a common pattern in studies focused on urban wildlife, with population and species relationships with urbanization varying due to underlying reasons such as the urban landscape, the presence of corridors, the dimensions and spatial arrangement of habitat patches (*Ortega-Álvarez & MacGregor-Fors, 2009*; *Fontana et al., 2011*) of tourism, and also cities with big parks used by both local people and tourists.

In our study, in Singapore we saw a slight increase in sampling effort when approaching the city center, but most of the occurrences are recorded at parks and gardens. Within a city, not all parks or green areas hold a continuum of values for VCF. However, in our data we find few gaps between dots, which reflects a good distribution of samplings for the VCF variable. In Colombo the urban matrix is imbricated with green avenues, parks and gardens. Therefore, the Asian water monitor appears in many parts of the city

homogenously, explaining its ability to reach particularly central parts of the city. A similar situation is observed in Jakarta, with higher distribution of observation ratio relying on the city center, and many of these observations occurring in gardens and water courses throughout the city. Bangkok demonstrates a similar pattern, with a smooth increase in the observation ratio as one approaches the city center and a tip around 6.25 km away from the border, and most of the records concentrated in small urban parks. In this case, *V. salvator* maybe uses the water canal system and even the sewage system to move within the city, disperse and connect parks and the outside of the city. Lastly, in Kuala Lumpur, where the city core is densely occupied by human-made buildings, the observation ratio tends to be similar even along the border-center gradient with a little increase in these core areas. Also, in this case, the ratio of observation in green areas is similar to that in the urban matrix. Similar studies in other reptiles, especially lizards, also contribute to explain how these animals exploit cities, with different selection according to the anthropogenic intensity and habitat availability observed within city boundaries. Thus, *Winchell et al. (2018)* show, in a study conducted in Puerto Rico, how *Anolis stratulus* tends to occur in more "natural" urban patches while *Anolis cristatellus* seems more attracted to human infrastructures. Moreover, *Dékány, Kövér & Babocsay (2015)* in their study of *Podarcis muralis* in Budapest (Hungary) also show higher densities in more diverse and with semi-natural elements urban patches. Regarding the data used, the lack of occurrences in certain areas might reflect different factors, such as cultural, low tourism, population density, lack of devices like smartphones to take pictures or even the legislation of the country itself that prevents their citizens from sharing data with the rest of the world (*Capdevila et al., 2020*; *Callaghan et al., 2021*; *Walker, Smigaj & Tani, 2021*). It should be also highlighted that citizen science is frequently subjected to representation biases that can affect spatial analysis or species modelling (*Kadmon, Farber & Danin, 2004*; *Franklin, 2010*; *Boria et al., 2014*), in such a way that observations accumulate near accessible paths, roads or in urban areas. In that sense, we realized that most of our observations are concentrated towards the city center in populated cities with high levels.

## CONCLUSIONS

In conclusion, our study highlights that *Varanus salvator*, despite being the second largest living lizard globally, and the largest lizard with established populations in urban areas, has drawn relatively little attention from ecologists regarding its colonization of numerous major cities across Asia. Thus, this species joins the list of those reptiles that successfully exploit cities (*Ackley et al., 2015*; *Davis & Doherty, 2015*; *Entiauspe-Neto, Perleberg & De Freitas, 2016*), and its example offers new insights to understand how reptiles interacts with urban environments. We show how the Asian water monitor occurs both in the urban matrix and big parks within cities, including those far from the city border, but in some cases also parks close to the sea, probably using water ecosystems to connect between urban areas and also with environments less influenced by the city. Further research is needed to disentangle the ecological aspects related to the urban life of the Asian water monitor along its distribution range. Specifically, the dispersal patterns and movements across the

urban matrix could help to explain both the colonization and the urban space exploited by the species. We suggest that an independent but probably non distant in time urban colonization, and not a single urban expansion in a leapfrog manner, is the most plausible option to explain the present case, considering the occurrence of the species in such distant cities, crossed by aquatic systems of different watersheds and habitats probably less suitable for the presence of the species. Such independent urban colonization pattern has been also demonstrated in other species (mostly birds) with notable dispersal capabilities and cities (*Evans et al., 2009*; *Mueller et al., 2018*). In this case, new studies including a genetic approach could contribute to elucidate whether our hypothesis is right or not. Moreover, a deeper explanation and comparison of their demographical parameters, the ecosystem services they provide as predators and scavengers (*Karunarathna et al. 2017*; *Luna, Romero-Vidal & Arrondo, 2021*), as well as the health status of urban individuals in comparison with their rural counterparts, could be relevant for the management of the urban populations, even more to consider the exploitation that the species suffer in less urbanized areas. Lastly, the relationship with tourists and citizens should be better assessed through social perception surveys, a key aspect to study in urban areas (*Botzat, Fischer & Kowarik, 2016*; *Ribeiro et al., 2021*). This is especially relevant in the case of the Asian water monitor, as they represent one of the most extreme cases of human-wildlife coexistence in urbanized areas (*Ceríaco, 2012*; *Pradhan & Yonle, 2022*).

## ACKNOWLEDGEMENTS

We acknowledge all the users of platforms as iNaturalist, whose contribution is offering a valuable source of global scientific information with growing scientific value. We also thank Jorge Juan Rueda for his contributions in the first steps of the project. Finally, we acknowledge Guillermo Fandos, Julio Rabadán and Pedro Romero Vidal for their useful opinions that helped us to improve the final version of this work, and also Nicholas West who contributed to improve the readability of the study. Finally, we thank the anonymous reviewers for the time and effort dedicated to reviewing our manuscript and the useful comments provided.

### Funding
The authors received no funding for this work.

### Competing Interests
The authors declare there are no competing interests.

### Author Contributions
- Álvaro Luna conceived and designed the experiments, performed the experiments, analyzed the data, prepared figures and/or tables, authored or reviewed drafts of the article, and approved the final draft.
- Armand Rausell-Moreno analyzed the data, prepared figures and/or tables, authored or reviewed drafts of the article, and approved the final draft.
## Data Availability

The data is available at GBIF:

- GBIF.org (27 November 2022) GBIF Occurrence Download https://doi.org/10.15468/dl.sg6x5g

- GBIF.org (11 December 2023) GBIF Occurrence Download https://doi.org/10.15468/dl.78mgfn.

## Supplemental Information

Supplemental information for this article can be found online at http://dx.doi.org/10.7717/peerj.17357#supplemental-information.

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
