# Peer review of "Unveiling the urban colonization of the Asian water monitor (Varanus salvator) across its distribution range using citizen science"

_PeerJ, doi:10.7717/peerj.17357_

## Round 0.1 · original submission · Major Revisions

· Academic Editor

Major Revisions

Thank you for submitting your manuscript to PeerJ and we look forward to receiving your revisions. This is an interesting and important study documenting the distribution range of a charismatic reptile as Varanus. The manuscript has potential and would surely be an interesting contribution to conservation within the framework of citizen science.

I will be happy to accept your paper pending major revisions, detailed by the referees - they are largely focused on clarifying several aspects of your work.

One of reviewers have major reservations about your manuscript, and the other reviewer has minor comments. Given this, I would like to see a major revision dealing with all of the comments. Please revise paying particular attention to the more critical comments. Please be aware that we consider these revisions to be major, and your revised manuscript will probably have to be re-reviewed.

Revisor 1 basically makes pointed comments within the text. The most important comments include including a new objective of the study and detailing some important points within Material and methods.

Reviewer 2, in Material and Methods, the reviewer suggests that you also carry out searches using other languages. It also suggests that they provide a new figure with a flowchart, showing the different steps for the selection such as the keywords used, the search engines, the number of results obtained, etc. On the other hand, it also suggests including all citations included in the information supplements.

I think this is very important information to record and report and I would certainly like to see it published.

**Language Note:** The review process has identified that the English language must be improved. PeerJ can provide language editing services - please contact us at [email protected] for pricing (be sure to provide your manuscript number and title). Alternatively, you should make your own arrangements to improve the language quality and provide details in your response letter. – PeerJ Staff

·

Basic reporting

no comment

Experimental design

I consider (commented in the manuscript) that an objective should be added that demonstrates the applicability of the survey of the data obtained. Highlight more the gaps in information that were obtained and the projections that arise with them.

Validity of the findings

no comment

Additional comments

Dear authors, those who are happy for the important work of collecting data for this species that is so "common" but so little used scientifically. I consider that his work provides good background information for possible future research, which, as discussed in his manuscript, he believes should be highlighted more. We also consider it very important that they highlight the information obtained from the platforms, especially those that favor and stimulate urban science, it is a good way to thank and encourage the continuity of the citizen's contribution. Congratulations again and my best wishes.

Reviewer 2 ·

Basic reporting

After reading the manuscript " Disentangling the overlooked urban colonization of
the Asian water monitor (Varanus salvator) along its
distribution range (#95666)" by Álvaro Luna and Armand Rausell-Moreno; I send my revision and opinion about it. My apologies to the authors if some commentaries could appear few polite, I am not a native English speaker.
First congratulations to the authors for conducting this interesting study on a charismatic Varanus lizard. The authors conducted a literature revision about Varanus salvator, searching those different studies made on its distributional range. Additionally, authors use different sighting points in urban environments, to understand the patterns of wildlife animal distribution and the urbanity expansion. In this sense, also, authors detected the presence of possible biases, which could affect negatively the knowledge and conservation of threatened animals. I consider that the manuscript presents potential and surely would be an interesting contribution to the conservation a citizen science framework. However, it needs some changes, which could enhance its present form.
Major issues detected are 1- Material and Methods, literature revision: A- Line 124- For the search, why the authors did not consider adding information from other languages? for example Spanish, Portuguese, Italian, and French? This information would be equally easy to read or interpret. I think that this could favour some biases in results. I suggest to the authors, that made also search employing another languages.
B- Lines 131-132 How many studies were discarded? I suggest to the authors that provide a new figure with a flux diagram, showing all the different steps for the selection: first key words used, searching motors used, how many results were obtained, the criteria used to discard some studies (also how many studies were discard), how many studies were analysed after this first selection, etc.
For an example of this, authors can consult the figure 1 from Ruiz-Monachesi & Martínez (Meta-analysis of Behavioural Research in Lizards Reveals that Viviparity Contributes Better to Animal Personality than Secretory Glands. Evol Biol (2023). https://doi.org/10.1007/s11692-023-09618-z)
2- Results: Line 206- 148 articles- Are all these studies used cited in the paper? I think these studies should get credit and should be cited accordingly. Please add to the list of references in your paper. You could even * the ones to indicate those used (i.e. 18 studies). I say this because often references thrown in the supplement are not picked up by search engines and this only reinforces the idea in the research community of meta-analyses being "parasites" (Gurevitch et al (2018) Nature, 555, 175–182; Nakagawa et al. 2020. Nature Ecology and Evolution, 4: 498-501).

After these issues, I upload a PDF reviewed with different and several minor issues and questions which I suggested will be clarified and incorporated before my recommendation of publication.
Please, there are several taxonomic names of species wrong written, without cursive, especially in the literature cited.

Experimental design

Material and Methods, literature revision: A- Line 124- For the search, why the authors did not consider adding information from other languages? for example Spanish, Portuguese, Italian, and French? This information would be equally easy to read or interpret. I think that this could favour some biases in results. I suggest to the authors, that made also search employing another languages.

Validity of the findings

The authors for conducting this interesting study on a charismatic Varanus lizard, the literature revision allows knowing the different study made on this peculiar lizard. Additionally, authors use different sighting points in urban environments, to understand the patterns of wildlife animal distribution and the urbanity expansion. In this sense, also, authors detected the presence of possible biases, which could affect negatively the knowledge and conservation of threatened animals. I consider that the manuscript presents potential and surely would be an interesting contribution to the conservation, a citizen science framework.

Annotated reviews are not available for download in order to protect the identity of reviewers who chose to remain anonymous.

---

## Round 0.2 · accepted · Accept

· Academic Editor

Accept

Thank you again for submitting your manuscript to PeerJ. This is an interesting and important study documenting the distribution range of a Varanus reptile. We received its revised version where we noted that each of the referees' comments were attended to. I evaluated the revision myself and am very satisfied with the current version which I consider ready for publication.